# Functional Role of STIM-1 and Orai1 in Human Microvascular Aging

**DOI:** 10.3390/cells11223675

**Published:** 2022-11-18

**Authors:** Mariam El Assar, Esther García-Rojo, Alejandro Sevilleja-Ortiz, Alberto Sánchez-Ferrer, Argentina Fernández, Borja García-Gómez, Javier Romero-Otero, Leocadio Rodríguez-Mañas, Javier Angulo

**Affiliations:** 1Fundación para la Investigación Biomédica, Hospital Universitario de Getafe, 28905 Getafe, Spain; 2Centro de Investigación Biomédica en Red de Fragilidad y Envejecimiento Saludable (CIBERFES), Instituto de Salud Carlos III, 28029 Madrid, Spain; 3Servicio de Urología, Hospital Universitario 12 de Octubre, Instituto de Investigación Sanitaria Hospital 12 de Octubre (imas12), 28041 Madrid, Spain; 4Fundación para la Investigación Biomédica, Hospital Universitario Ramón y Cajal, 28034 Madrid, Spain; 5Servicio de Histología-Investigación, Unidad de Investigación Traslacional en Cardiología (IRYCIS-UFV), Hospital Universitario Ramón y Cajal, 28034 Madrid, Spain; 6Servicio de Geriatría, Hospital Universitario de Getafe, 28905 Getafe, Spain

**Keywords:** aging, vascular function, Orai channel, STIM-1, human mesenteric arteries

## Abstract

The impact of aging on vascular function is heterogeneous depending on the vascular territories. Calcium regulation plays a key role in vascular function and has been implicated in aging-related hypercontractility of corpus cavernosum. We aimed to evaluate stromal interaction molecule (STIM)/Orai system involvement in aging-related vascular alterations in the human macro and microvasculature. Aortae specimens and mesenteric arteries (MA), obtained from 45 organ donors, were functionally evaluated in organ chambers and wire myographs. Subjects were divided into groups either younger or older than 65-years old. The expressions of STIM-1, Orai1, and Orai3 were determined by immunofluorescence in the aorta and MA, and by Western blot in the aorta homogenates. The inhibition of STIM/Orai with YM-58483 (20 μM) reversed adrenergic hypercontractility in MA from older subjects but did not modify aging-related hypercontractility in the aortic strips. Aging was related to an increased expression of Orai1 in human aorta, while Orai1 and STIM-1 were upregulated in MA. STIM-1 and Orai1 protein expressions were inversely correlated to endothelial function in MA. Circulating levels of Orai1 were correlated with the inflammatory factor TNF-α and with the endothelial dysfunction marker asymmetric dimethylarginine. Aging is associated with an increased expression of the STIM/Orai system in human vessels with functional relevance only in the microvascular territory, suggesting its role in aging-related microvascular dysfunction.

## 1. Introduction

Aging is the main risk factor for cardiovascular disease (CVD), even in the absence of traditional risk factors, while CVD is considered to be the principal contributor to morbidity and mortality in older populations [1]. Interestingly, the aging process is associated with both structural and functional alterations at the vascular level, leading not only to an increase in cardiovascular events in older subjects, but also to functional decline, cognitive deterioration, and frailty [2]. Endothelial dysfunction is one of the phenotypic characteristics of cardiovascular alterations related to aging [3]. In fact, different evidence has clearly demonstrated the presence of endothelial dysfunction, manifested by impaired endothelium-dependent vasodilation, associated with the aging process in the macro and microvasculature of animal models and humans [4,5,6]. Although the underlying mechanisms of vascular dysfunction related to the aging process are not completely known, oxidative stress and inflammation stand out as two main candidates leading to vascular impairment in aging [7]. However, aging is also related to another vascular alteration that may play a role in enhanced CVD risk such as increased agonist-induced vasoconstriction [8,9]. It is of importance to note that the impact of aging on vascular function is rather heterogeneous depending on the vascular territory evaluated [9,10,11].

Changes in cellular Ca^2+^ levels play a crucial role in vascular function and blood pressure regulation [12]. Importantly, vascular calcium homeostasis is disturbed in different pathological situations [13] and in aging [14,15]. Different systems control the cell’s capacity to manage the calcium level, where the store-operated calcium entry (SOCE) through stromal interaction molecule (STIM)-gated Orai channels plays a key role [16]. In addition to their role in Ca^2+^ signaling, STIM and Orai have been shown to participate in the regulation of metabolism and mitochondrial function, and their activities seems to be susceptible to redox modifications [17]. The STIM/Orai system is essential for cellular homeostasis and its disruption is linked to various diseases associated with aging such as CVD [17]. In line with this, it has been suggested that STIM/Orai channels participate in the pathophysiology of vascular disease, independent of their contribution to SOCE [18,19]. Moreover, augmented activation of the STIM/Orai system has been suggested to play a role in increased basal tonus and vascular reactivity in hypertensive rats [20,21].

Aging is known to lead to vascular dysfunction, in part mediated by altered Ca^2+^ homeostasis and signaling in smooth muscle cells, as well as endothelial cells [12]. However, the scarce evidence that exists does not allow for obtaining a consistent idea about the influence of aging on the STIM/Orai system in the vasculature. In line with this, SOCE-induced contractions have been observed to increase in mesenteric arteries of old rats, while they were reduced in the aorta of the same animals. This was accompanied by a decreased expression of Orai1 in the aortic tissue with aging, and an increased expression of Orai1 and decreased STIM-1 expression in the mesenteric arteries from old rats [22]. Furthermore, we recently described an enhanced contribution of STIM/Orai signaling to aging-related hypercontractility in human corpus cavernosum and penile arteries and in the corpus cavernosum of old rats, accompanied by augmented immunodetection of the Orai3 channel in these tissues [23]. This evidence points to a heterogeneous role of the STIM/Orai system in vascular aging among specific vascular territories. However, the impact of physiological aging on the STIM/Orai system in other human vascular territories has not been previously established. Therefore, the main objective of the current study was to evaluate the role of the STIM/Orai system in vascular alterations associated with aging in human macrovasculature (aortic strips) and microvasculature (small mesenteric arteries) obtained from younger and older organ donors.

## 2. Materials and Methods

### 2.1. Human Tissues

Human aorta and epiplon, in addition to blood samples, were obtained from 45 deceased organ donors at the moment of organ transplantation. Written informed consent was provided by donors’ relatives, including specific research informed consent for the tissue procurement. Ethical approval of the study protocol and informed consent were obtained from the Ethics Committees of the Hospital Universitario de Getafe (Ethics Approval procedure A06/15, 30 April 2015); the Hospital Universitario Doce de Octubre (Ethics Approval procedure 16/045, 25 February 2016); and the Hospital Universitario Ramón y Cajal, Madrid, Spain (Ethics Approval procedure 16/045, 3 March 2016). The tissues were collected in sterilized M-400 solution (composition per 100 mL: 4.19 g of mannitol; 0.205 g of KH_2_PO_4_; 0.97 g of K_2_HPO_4_•3H_2_O; 0.112 g of KCl; and 0.084 g of NaHCO_3_; pH 7.4) and were maintained at 4–6 °C. For the experimental assessments, the specimens were transported under the same conditions to the Getafe and Ramón y Cajal University Hospitals. Time elapsed from extraction until processing and functional evaluation ranged between 16 and 24 h, a time during which the vascular specimens were maintained as being viable [9,23]. The clinical characteristics of the participants are depicted in Table 1.

Exclusion criteria were applied at the stage of tissue collection. Subjects presenting any infectious disease were excluded from the study. All of the tissues were evaluated when they were delivered in time to the laboratories under adequate storage conditions. Functional and molecular evaluations were performed on the same subject when a sufficient amount of vascular tissue was obtained. Tissues derived from organ donors were divided into two groups: younger than 65-years old (<65 group) and older than 65-years old (>65 group).

### 2.2. Functional Evaluation of Human Mesenteric Arteries (MA)

Human mesenteric small arteries (internal diameter (mean ± SEM): <65-years old: 347.1 ± 25.5 μm, *n* = 16; and >65-years old: 398.9 ± 79.6 μm, *n* = 9, *p* > 0.05) were carefully dissected from the epiplon specimens. Arterial ring segments about 2 mm long were mounted on a small vessel myograph (Danish MyoTechnology, Aarhus, Denmark) for isometric tension recordings, as previously described [9,24,25]. Vascular segments were equilibrated in a Krebs–Henseleit solution (KHS) for 30 min. The composition of the KHS solution comprised (mM) NaCl 119, KCl 4.6, CaCl_2_ 1.5, MgCl_2_ 1.2, NaHCO_3_ 24.9, glucose 11, KH_2_PO_4_ 1.2, and EDTA 0.027. Arterial segments were maintained at 37 °C and continuously bubbled with a mixture of 95% O_2_ and 5% CO_2_ in order to obtain a pH of 7.4. The arteries were then set to the 90% of the determined internal circumference under a transmural pressure of 100 mmHg (L_100_), at which point the force development was close to maximal. Consequently, vessel preparations were exposed to 125 mM K^+^ (KKHS, equimolar substitution of NaCl for KCl in KHS) in order to assess their viability and the contractile response was analyzed. Vessel preparations were washed with KHS and, after a stabilization period, contractile responses in MA were evaluated by cumulative additions of norepinephrine (NE, 1 nM to 10 μM) to the chambers. Concentrations were added in semi-logarithmic increases at 3 min intervals or when the contraction to the previous concentration was stabilized. YM-58483 (Tocris, Bristol, UK) (20 μM) or the vehicle (DMSO) were added 30 min before the concentration–response curves started. The concentration of YM-58483 was selected based on previous experience [23,26]. The relaxation response was evaluated in arterial segments precontracted with the thromboxane analogue, U46619 (Sigma-Aldrich, St Louis, MO, USA) (0.1–0.3 μM), through cumulative additions of YM-58483 (0.1–30 μM) to the chambers.

For evaluation of the endothelial function in MA specimens, arterial segments were precontracted with U46619 (0.1–0.3 μM) and exposed to cumulative increasing concentrations of bradykinin (Sigma-Aldrich) (BK, 10 nM to 3 μM). Vascular segments failing to relax more than 10% to the highest BK concentration were discarded. The pEC_50_ values for each subject were calculated as the negative log of the BK concentration required in order to obtain 50% relaxation and were considered as the indicator of endothelial function.

### 2.3. Functional Evaluation of Human Aorta

Aortic segments derived from the organ donors were cut into transversal strips of about 7 mm in length and 2 mm in width, and were immersed in 8 mL organ chambers that contained KHS at 37 °C and were continuously bubbled with a mixture of 95% O_2_ and 5% CO_2_, in order to maintain a pH of 7.4 [9]. After an equilibration period of 90 min at 1.5 g of tension, the aortic strips were exposed to a high K^+^ concentration (KKHS) and the contractile response was measured. Contractile responses to accumulative additions of NE (1 nM to 100 μM) were evaluated in the same way as described for MA. STIM/Orai inhibitor YM-58483 (20 μM), or the vehicle (DMSO) were added 30 min before the concentration–response curves started. The relaxation response was evaluated in the aortic strips precontracted with NE (1–10 μM) through cumulative additions of YM-58483 (0.1–30 μM) to the chambers.

### 2.4. Immunofluorescence Assay

Freshly isolated MA and aortae specimens were immersed in increasing percentages of saccharose (10–30% *w*/*v*), embedded in an optimal cutting temperature compound (OCT; Sakura Finetek, Tokyo, Japan), and kept at −80 °C until the immunofluorescence assays were carried out, as described previously [23]. Then, 6 μm sections were obtained from the OCT blocks by cutting in a cryostat. After removing the OCT, fixing the tissues, and removing the autofluorescence with acetone and methanol, the sections were incubated with rabbit antibodies against STIM-1, Orai1, and Orai3 (1:200 dilution; Novus Biologicals, Littleton, CO, USA) overnight at 4 °C. After washout in phosphate buffered saline plus 0.3% Triton X-100, the sections were incubated with a secondary Alexa Fluor 488-conjugated goat anti-rabbit antibody (dilution 1:250; Life Technologies, Alcobendas, Spain) and with diamidino-2-phenylindole (DAPI; Life Technologies) to counterstain the nuclei for 1 h at room temperature. The sections were mounted and viewed using fluorescence microscopy (Olympus BX51, Olympus Corporation, Tokyo, Japan). Controls without primary antibodies showed no unspecific reactivity. Five random images from each specimen were captured, and the fluorescence intensity was quantified and normalized with the nuclei number by using Image J 1.48i software (McBiophotonics Image J, NIH, Bethesda, MD, USA). An average value for each specimen was obtained. The group corresponding to each specimen was blinded for the investigator capturing and quantifying the immunofluorescence images.

### 2.5. Western Blot Analysis

Western blot analyses were carried out as previously described [23,26]. Aortic tissue samples derived from organ donors were flash frozen in liquid nitrogen and kept at −80 °C until the proteins were extracted. For obtaining the total protein extracts, aortic tissue was homogenized in a T-PER lysis buffer (Pierce Biotechnology, Inc., Rockford, IL, USA) according to the manufacturer’s instructions, and 1× of Protease Inhibitor Cocktail (Roche Diagnostics, Indianapolis, IN, USA) was added. A total of 15 μg of protein extracts were separated by SDS-PAGE on a 10% polyacrylamide gel. The proteins were transferred to PVDF membranes and blocked for 5 min with an EveryBlot blocking buffer (Bio-Rad, Hercules, CA, USA). The membranes were incubated overnight at 4 °C with a specific rabbit antibody against STIM-1 (Novus, Littleton, CO, USA, cat.# NBP1-52849, dilution 1:1000), mouse antibody against Orai1 (ThermoFisher Scientific, Waltham, MA, USA, cat.# MA5-15776, dilution 1:500), rabbit antibody against Orai3 (ThermoFisher Scientific, cat.# PA5-22273, dilution 1:500), and a mouse antibody against β-actin (Novus, cat.# NB600-501, dilution 1:5000), which was used as the loading control. Consequently, the membranes were incubated with goat anti-mouse (1:5000 dilution; Novus, cat.# NBP2-30347H) or goat anti-rabbit horseradish peroxidase-conjugated secondary antibody (1:10,000 dilution; Novus, cat.# NB7160) for 1 h at room temperature. The blots were visualized by the ECL detection system (ThermoFisher Scientific). The results were quantified by densitometry, using QuantityOne/Chemi-Doc 6.0 Software (Bio-Rad, Barcelona, Spain).

## 3. Results

### 3.1. Orai Channel Inhibition Reverses Age-Related Adrenergic Hypercontractility of Mesenteric Arteries but Has No Functional Effect on the Aorta from Older Human Subjects

Despite not detecting significant differences in contractile responses to a high concentration of K^+^ in human mesenteric arteries (MA) and in human aorta associated to age (mean ± SEM in MA: <65-years old: 7.97 ± 1.07 mN, *n* = 16 vs. >65-years old: 7.30 ± 3.02 mN, *n* = 9, *p* > 0.05; in aorta: <65-years old: 1.38 ± 0.30 g, *n* = 6 vs. >65-years old 1.12 ± 0.33 g, *n* = 6, *p* > 0.05), the addition of cumulative concentrations of NE to the organ bath evoked a concentration-dependent contraction that was significantly increased in MA (Figure 1A) and aortic strips (Figure 1B) derived from older subjects (>65-years old) when compared with those obtained from younger subjects (<65-years old). Acute treatment of MA for 30 min with the Orai channel inhibitor YM-58483 (20 μM) inhibited adrenergic-induced contractions in both of the of age evaluated groups (younger and older subjects), although the magnitude of inhibition was more marked in the older subject group (−32.83 ± 6.10% (*n* = 16) vs. −68.21 ± 20.15% (*n* = 9), *p* < 0.05 in % E_max_ to NE for younger and older subjects, respectively). Moreover, no differences were detected between the <65-year-old and >65-year-old group after inhibition with YM-58483 (Figure 1A). However, although the aortic strips presented an age-related adrenergic hypercontractility, Orai channel inhibition with YM-58483 (20 μM) did not modify these responses in the aortic strips derived from older subjects. In addition, YM-58483 had no significant effect on the contractions elicited by NE in the aorta from younger subjects (Figure 1B). These results seem not to be conditioned by the inclusion of both male and female subjects in the analysis as the results did not change when only analyzing the responses in the MA and aortic strips from female subjects older than 65-years old. In this sense, when considering only female subjects, Orai inhibition caused a marked reduction of NE-induced contractions in the microvessels, but did not cause significant effects in the aorta (Appendix A), in a similar way to that observed when considering all of the subjects.

As the prevalence of diabetes mellitus and hypertension was significantly increased in the older subject group, we further analyzed the effect of the STIM/Orai inhibitor on the contractile responses induced by NE in the >65-year-old group only considering those subjects without diabetes (Appendix A). The same was done when analyzing the responses in the older group only considering those subjects without hypertension (Appendix A). Interestingly, YM-58483 significantly inhibited adrenergic contractions in both of the groups of subjects evaluated (E_max_ for NE: >65 without diabetes: 152.70 ± 17.98%, *n* = 6 vs. 58.87 ± 11.27%, *n* = 6 for >65 without diabetes + YM-58483, *p* < 0.05; >65-years old without hypertension: 133.80 ± 8.57%, *n* = 3 vs. 72.57 ± 17.62% for >65 without hypertension + YM-58483, *n* = 3, *p* < 0.05), reinforcing the concept that the role of the STIM/Orai system in the hypercontractility of aged MA is not dependent on the increased presence of hypertension or diabetes with aging.

Furthermore, the accumulative addition of YM-58483 (0.1 μM–30 μM) to pre-contracted vascular segments provoked a concentration-dependent vasodilation in the MA, but this relaxation was almost absent in the aortic strips (Figure 1C). This observation can also be seen in representative tracings of YM-induced relaxations in the MA and aorta (Figure 1D,E). The quantification of YM-induced relaxations showed significant differences between the two types of vascular vessels (Figure 1C).

### 3.2. Orai1 Expression Is Increased in Aorta Derived from Aged Subjects

Western blot was performed to assess the expression of STIM-1, Orai1, and Orai3 in the aorta homogenates (Figure 2A). As illustrated in Figure 2C, the quantitative expression of Orai1 was significantly increased in the aortae from older subjects in comparison with those from younger subjects. In contrast, no significant changes in STIM-1 (Figure 2B) and Orai3 expression (Figure 2D) were detected in the older subjects vs. the younger ones. Supportive information for these assays is provided in the Appendix A, including a comparison of the β-actin expression between the younger and older subjects (Appendix A), raw data confirmation of Orai1 overexpression in the older subjects (Appendix A), and a representative complete immunoblot for Orai detection in the tissue homogenates from the aortic strips from the younger and older subjects (Appendix A). The increase in Orai1 protein expression in the aorta was further confirmed through immunofluorescence detection. Orai1 was clearly up-regulated in the aortic sections derived from older subjects when compared with the younger ones (Figure 2E,F). When the Orai1 expression was represented against each subject’s age, a significant and a positive association was detected (r^2^ = 0.325, *p* < 0.0001) (Figure 2G). Furthermore, age was significantly associated with adrenergic contraction, determined as the maximum response (E_max_) to NE (Figure 2H). In contrast, no association was observed between Orai1 and adrenergic vasoconstriction (E_max_ to NE) in the donor’s aorta (r^2^ = 0.060, *p* > 0.05) (Figure 2I).

### 3.3. Human Aging Is Associated with an Increased Expression of STIM-1 and Orai1 in MA

STIM-1 and Orai1 proteins were immunodetected in human mesenteric arteries (Figure 3A,D). Weak Orai3 protein immunodetection was observed in MA from subjects <65-years old and in sections derived from the older subjects (Figure 3E,F). The quantification of the fluorescence signal in MA showed a significant increase in STIM-1 (Figure 3G) and Orai1 (Figure 3H) expression in the vessels from older subjects when compared with the arteries derived from the younger ones. In contrast, Orai3 immunodetection was not significantly modified by age, as no significant differences were observed between younger and older subjects (Figure 3I).

In addition, a potential relationship between the STIM/Orai expression and endothelial function in MA was evaluated. In this sense, when individual relative immunofluorescence values of STIM-1, Orai1, and Orai3 proteins were plotted against endothelium-dependent vasodilation, represented as pEC_50_ for the BK of each subject, a significant negative correlation was obtained for the STIM-1 (r^2^ = 0.335, *p* < 0.05) and Orai1 expression (r^2^ = 0.419, *p* < 0.05), but not for Orai3 (r^2^ = 0.109, *p* > 0.05) (Figure 4A,C).

### 3.4. Plasma Orai1 Concentrations Are Associated with Age-Related Circulating Markers of Endothelial Dysfunction and Inflammation

To assess the possible association of Orai1 with markers of endothelial dysfunction and inflammation, which are closely related to the aging process, the plasma concentrations of Orai1, asymmetric dimethylarginine (ADMA), and tumor necrosis factor-α (TNF-α) were determined in the samples derived for organ donors. Concentrations of Orai1 in donors’ plasma were related to endothelial dysfunction, as suggested by the positive and significant correlation of these concentrations with ADMA levels (Figure 5A). In a similar way, circulating Orai1 was positively correlated with plasma concentrations of the inflammatory factor TNF-α (Figure 5B).

## 4. Discussion

The results of the present study show, for the first time, that Orai1 is a potential marker of vascular functional alterations associated with human aging in the microvasculature. This is supported by the fact that the inhibition of the STIM/Orai system reduced significant aging-related hypercontractility in small mesenteric arteries. This functional relevance is further associated with an increase in STIM-1 and Orai1 channel subtype expression in small mesenteric arteries from older subjects, which were both negatively correlated with impaired endothelium-dependent vasodilation. In addition, the plasmatic concentration of Orai1 was positively associated with age-related markers of endothelial dysfunction and inflammation. Interestingly, although Orai1 was upregulated in the aortic strips derived from older organ donors, no functional relevance of this system was evidenced.

The impact of aging on vasculature plays a prominent role in the morbidity and mortality of older subjects. In fact, aging is considered the principal cardiovascular risk factor [27]. Endothelial dysfunction is considered one major age-related phenotype that is responsible for CVD development [28]. Consistent evidence has reported impaired endothelium-dependent relaxation in the macro and microvasculature of aged animal models and of older adults [4,5,6,29]. Importantly, an altered vessel contraction function is also present in aging [22]. In line with this, the detected adrenergic hypercontractility in the mesenteric arteries and aorta derived from old organ donors is consistent with previous reports in humans and aged rats [8,9,10,23]. The increase in contractions elicited by norepinephrine was not accompanied by significant alterations in contractile responses to a high concentration of potassium, which probably suggests that aging does not seem to modify this specific type of contractile response. This is in agreement with previous observations [9] and allows for using K^+^-induced contractions to normalize agonist-induced contractions.

Vasoconstriction is triggered by an increase in the intracellular-free calcium concentration [30]. One of the most ubiquitous regulated means of Ca^2+^ influx into cells is the store operated Ca^2+^ entry (SOCE) pathway mediated by STIM and Orai proteins [18]. Compelling evidence supports the idea that SOCE pathway dysregulation seems to play an important role in the development of vascular alterations [31], including cellular remodeling [32] and increased vascular contractility [20]. Furthermore, one of the prominent changes of aging is the effect on Ca^2+^ signaling [33]. In this sense, Rubio et al. reported increased phenylephrine-induced contractions in mesenteric arteries, which was accompanied with abnormal calcium handling in aged rats [34]. Importantly, very little is known regarding how aging impacts STIM/Orai activity at a vascular level, as well as its key role in aging-related vascular alterations. In line with this, an increased contribution of STIM/Orai signaling to aging-related hypercontractility together with Orai3 upregulation in the corpus cavernosum and penile resistance arteries derived from older subjects has recently been reported. These observations were extended to the corpus cavernosum of aged rats [23]. Despite not evaluating the same vascular bed, similar results were observed in our study regarding the enhanced contribution of STIM/Orai to hypercontractility associated with aging in the human microvasculature. In fact, the antagonism of STIM/Orai significantly and completely inhibited age-related hypercontractility in small mesenteric arteries, as evidenced by the magnitude of the inhibitor effect exerted by YM-58483 in vessels derived from subjects older than 65 when compared with vessels obtained from younger subjects (Figure 1A). Moreover, the antagonism of the STIM/Orai system reveled that Orai channels contributed to aging-related hypercontractility, regardless of the presence of diabetes and hypertension conditions in the sample (Appendix A). Furthermore, and supporting functional evidence, the addition of the YM-58483 produced concentration-dependent relaxation in the precontracted MA but not in the aortic strips (Figure 1C). In contrast, in the aortic strips, despite presenting an enhancement of norepinephrine-induced contraction, YM-58483 had no effect on these responses, suggesting the absence of functional relevance of STIM/Orai signaling in the contractility of this specific type of tissue (Figure 1B). In addition, these results support that the impact of aging on calcium signaling is largely different in the mesenteric arteries and aorta, which is consistent with previous evidence regarding aged rats [22], and supports the previously described heterogeneous impact of aging on vascular function depending on vascular territory [9,10].

The role of the STIM/Orai signaling system in enhanced agonist-induced hypercontractility related to age is further supported by the impact of aging on the expression of STIM/Orai proteins in small mesenteric arteries (Figure 3). Interestingly, expression levels of the Ca^2+^ sensor STIM-1 and calcium channel subtype Orai1 were increased in this vascular territory, as revealed by the immunofluorescence assays. An increased Orai1 expression is consistent with previous evidence regarding the mesenteric arteries of aged rats, but no increase in the STIM-1 expression was observed in contrast with our human evidence [22]. The latter reflects that the impact of aging is different according to the species evaluated and the importance of validating the findings obtained from animal models in humans. STIM-1 and Orai1 are the elements of the STIM/Orai system most involved in vascular physiology and pathophysiology. However, while some evidence points to the involvement of Orai3 in cardiovascular pathophysiology, there is scarce evidence on the relevance of Orai2 in vascular physiology/pathophysiology, and its role needs to be determined [17]. Furthermore, we previously reported that Orai3 was overexpressed in the human corpus cavernosum in physiologic aging [23], while both Orai1 and Orai3 were upregulated in the cavernosal tissue from men with erectile dysfunction [26]. Based on this previous evidence, we focused in the assessment of STIM-1, Orai1, and Orai3 expressions in human micro and macrovasculature in the context of aging. Our results point to STIM-1 and Orai1 upregulation as the main alteration in the STIM/Orai system with aging in mesenteric microvessels.

Importantly, STIM-1 and Orai1 relative fluorescences were significantly associated with another major characteristic of vascular aging, impaired endothelium-dependent vasodilation, which is considered the harbinger of cardiovascular disease development [28], as supported by simple regression analyses in the human mesenteric arteries (Figure 4). This suggests that there is an association between the aging-related increase in the STIM/Orai vascular expression and the development of microvascular dysfunction. With respect to the potential association of Orai inhibition with endothelial function, some considerations should be taken. Fine mutual regulation of calcium activated potassium channels (K_Ca_) and calcium channels has been proposed. K_Ca_ are activated by Ca^2+^. This activation promotes hyperpolarization of excitable cells, such as vascular smooth muscle cells, closing voltage-gated calcium channels and thus reducing intracellular Ca^2+^. However, K_Ca_ also activates non-voltage gated Ca^2+^ channels (such as Orai), mainly in non-excitable cells such as endothelial cells [35]. Thus, the Orai–K_Ca_ interaction could represent both negative and positive feedback loops. K_Ca_ participates in endothelium-derived hyperpolarizing factor (EDHF)-mediated vasodilation [36], which accounts for endothelial vasodilation in small size arteries such as human mesenteric microvessels [37,38]. Potentially, the reduction of intracellular calcium by Orai inhibition could result in reduced K_Ca_ activation and impaired EDHF-mediated vasodilation. This seems to not be an important mechanism in our preparations as a lower expression of Orai1 in mesenteric arteries is related to better endothelial vasodilation. However, further research directed to evaluate the influence of STIM/Orai inhibition on endothelial function and, specifically on EDHF-mediated vasodilation, is warranted.

Interestingly, although the STIM/Orai inhibitor had no effect on aging-related hypercontractility in aortic specimens, a significantly increased expression of Orai1 was detected in aortic homogenates derived from older subjects. The latter was further confirmed by immunofluorescence assays. Furthermore, although the Orai1 expression was positively related to increased age, and despite observing a significant correlation between increased contraction to adrenergic stimulation and age, Orai1 was not associated with the hypercontractility observed in this specific vascular tissue (Figure 2). This points to the fact that the parallel increases in the aortic Orai1 expression and NE-induced contractions with aging are not functionally related. It is of importance to note that although the STIM/Orai calcium entry system does not seem to have functional relevance in age-related hypercontractility in this macrovascular territory, we cannot discard a possible pathophysiological role of this system. In fact, upregulation of the STIM/Orai system in vascular pathophysiology has been mostly associated with an increase in vascular proliferation and remodeling [32,39], which are features of macrovascular aging [40].

Taking into account that it is not easy to determine vascular function in a clinical scenario, the exploration and search for novel biomarkers that reflect vascular performance have attracted much interest. In line with this, we evaluated the association between the plasmatic concentrations of Orai1 and two known biomarkers, ADMA and TNF-α, involved in vascular aging [7]. In line with this, higher plasmatic concentrations of ADMA, an endogenous inhibitor of the endothelial nitric oxide synthase, and of TNF-α, a biomarker of inflammation that plays an important role in endothelial dysfunction and in the development and progression of atherosclerosis [41], have been previously observed to be related to impaired endothelial function in humans [9]. In the present study, we detected a significant and positive association between Orai1 circulating levels with ADMA levels or with TNF-α in organ donors (Figure 5). Taken together, these results suggest that Orai1 may represent a potential biomarker of vascular alterations with aging.

## 5. Conclusions

In this paper, we show for the first time that aging-induced hypercontractility is related to functional enhancement of the STIM/Orai calcium entry system in the human microvasculature, as evidenced by a significant reduction in adrenergic-induced contractions in the presence of the STIM/Orai inhibitor in small mesenteric arteries derived from aged organ donors. In addition, STIM-1 and Orai1 protein expressions were upregulated in small arteries and were negatively correlated with endothelial vasodilation. Furthermore, Orai1 was positively associated with markers of inflammation and endothelial dysfunction that are closely related to age. In contrast, despite detecting an increase in Orai1 expression in aortic strips derived from aged subjects, no functional relevance for STIM/Orai was observed in the contractility of this large vessel. These results suggest a specific role of the STIM/Orai system in aging-related microvascular dysfunction.

## Figures and Tables

**Figure 1 cells-11-03675-f001:**
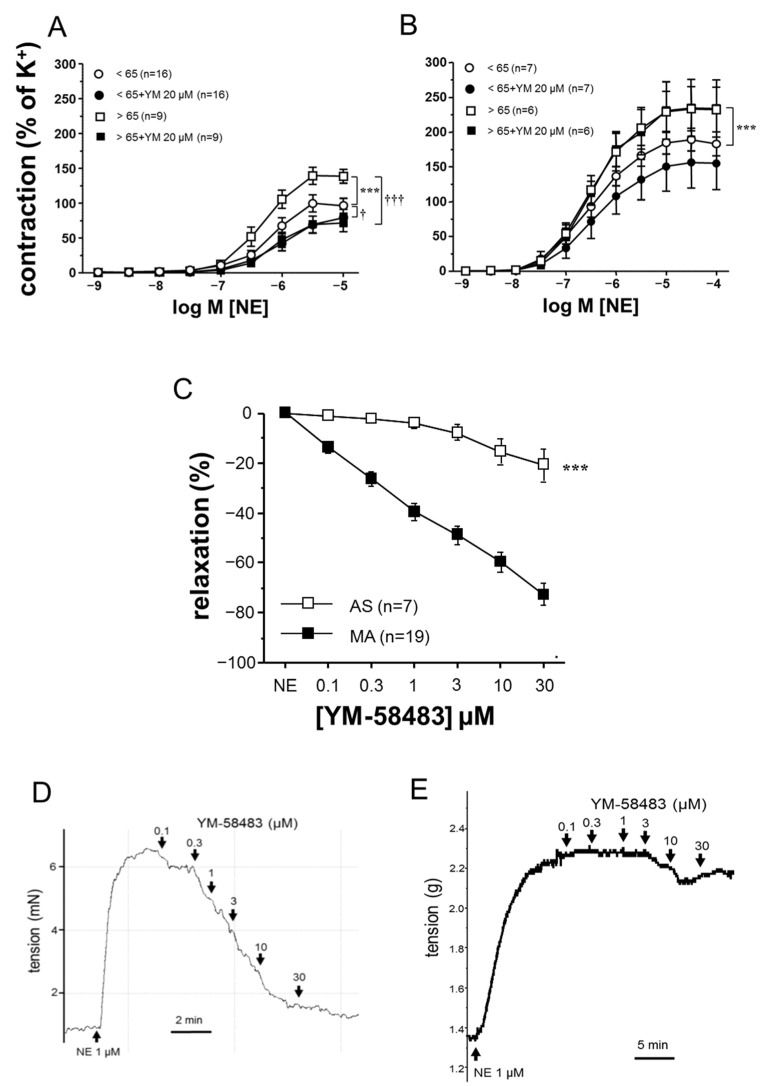
Aging-related hypercontractility to adrenergic stimulation in the human mesenteric artery is reversed by Orai channel inhibition. Effects of Orai channel inhibition with YM-58483 (20 μM) on norepinephrine (NE)-induced contraction in human mesenteric arteries (MA (**A**)) and human aortic strips (AS (**B**)) obtained from organ donors <65-years and >65-years old. Data are expressed as mean ± SEM of the percentage of contraction induced by 125 mM K^+^ before the addition of YM-58483 or the vehicle (0.2% DMSO). *n* indicates the number of subjects. *** *p* < 0.001 vs. <65-years old; ^†^
*p* < 0.05, ^†††^
*p* < 0.001 vs. respective responses without YM-58483 by two-factor ANOVA followed by Bonferroni’s correction. (**C**) Complete concentration–response curves of YM-58483-induced relaxation in human AS and MA precontracted with NE obtained from organ donors. Data are expressed as mentioned above. *** *p* < 0.001 vs. MA. Representative tracings of these responses in MA and AS are shown in (**D**,**E**), respectively.

**Figure 2 cells-11-03675-f002:**
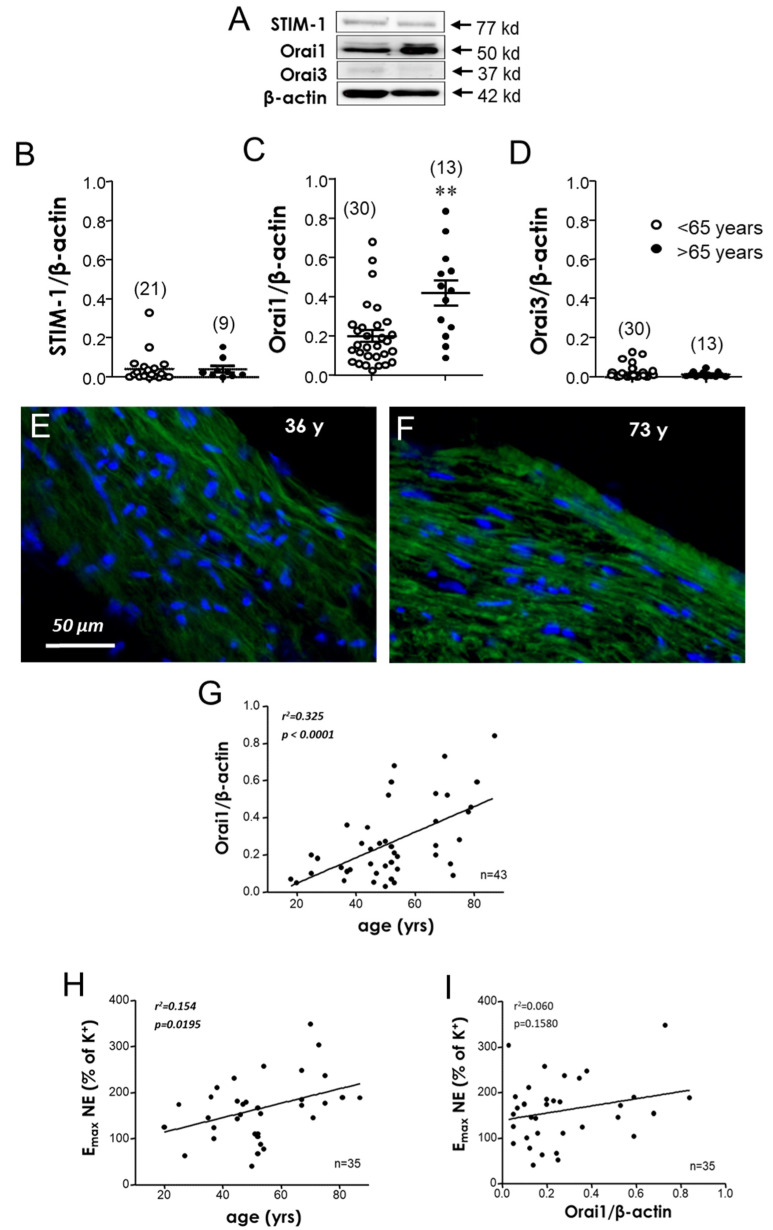
Aging is associated with an overexpression of the Orai1 channel in human aortic strips. (**A**) Representative immunoblots for the detection of STIM-1, Orai1, and Orai3 and corresponding β-actin in aortic strip homogenates from subjects <65-years and >65-years old. Quantification of expression assays for STIM-1, Orai1, and Orai3 are displayed in panels (**B**–**D**), respectively. Data are expressed as mean ± SEM of band intensities normalized by respective β-actin band intensities. *n* indicates the number of subjects. ** *p* < 0.01 vs. <65-year-old group by unpaired *t*-test. (**E**,**F**) Representative immunofluorescence images for the detection of Orai1 (green fluorescence) in cryosections of aortic strips from a 36 year-old subject and a 73 year-old subject, respectively. Nuclei are counterstained in blue. Magnifications: ×200. (**G**) Simple regression analysis of individual values of the Orai1 expression in human aortic strip homogenates with respect to the age of the subject from which the tissue was obtained. (**H**) The association between adrenergic contraction determined as the maximum response (E_max_) to norepinephrine (NE) expressed as the percentage of K^+^-induced contraction with respect to the age from the same subject. (**I**) Association between aortic Orai1 quantification and E_max_ to NE. Every point represents the averaged E_max_ value of the strips from one single subject. Coefficients of determination (r^2^) and *p* values are indicated for each association (in ***bold italic*** when significant).

**Figure 3 cells-11-03675-f003:**
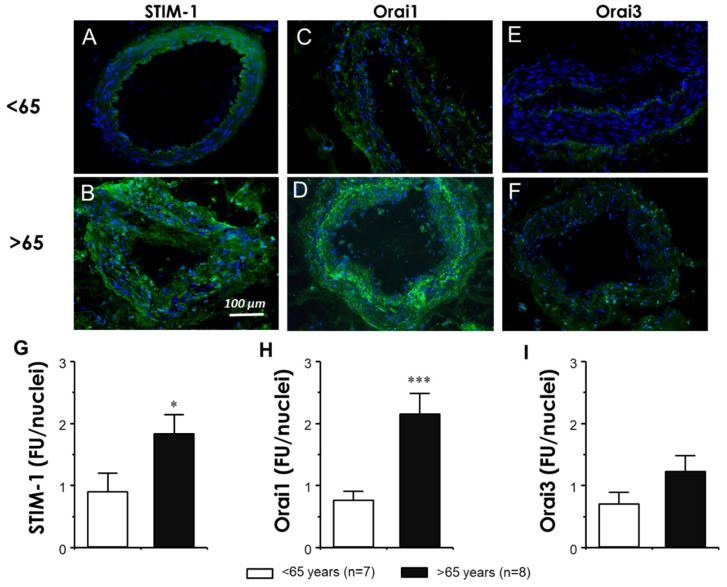
STIM-1 and Orai1 are up-regulated in human aged mesenteric arteries. Upper panel shows the representative immunofluorescence images for detection (green fluorescence) of STIM-1 (**A**,**B**); Orai1 (**C**,**D**) and Orai3 (**E**,**F**) in cryosections of mesenteric arteries from organ donors younger than 65-years old (**A**,**C**,**E**) and older than 65-years old (**B**,**D**,**F**). Nuclei are counterstained in blue. Magnifications: ×100. Quantification of expression assays for STIM-1, Orai1, and Orai 3 are displayed in panels (**G**–**I**), respectively. Data are expressed as mean ± SEM of STIM/Orai arbitrary units of fluorescence intensities normalized by number of nuclei. *n* indicates the number of subjects from which the tissues were obtained for the experiments. * *p* < 0.05, *** *p* < 0.001 vs. <65-year-old group by unpaired *t*-test.

**Figure 4 cells-11-03675-f004:**
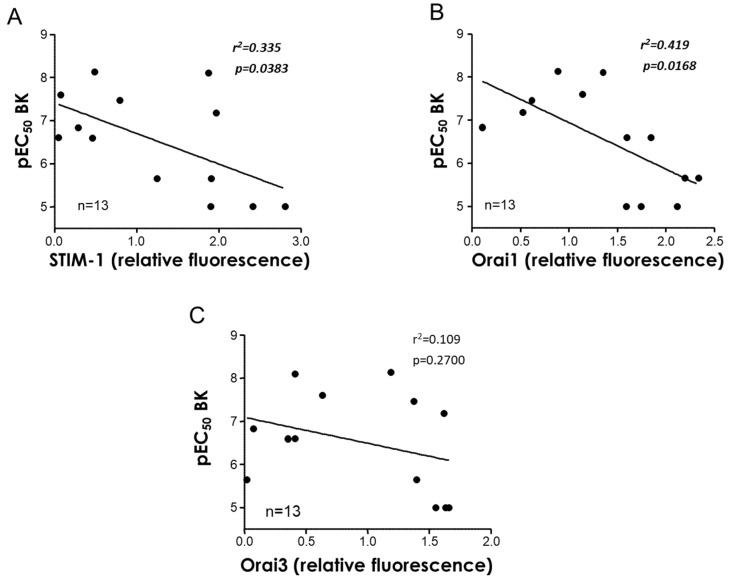
STIM1 and Orai1 are associated with vascular function in human mesenteric arteries. Simple regression analysis for STIM-1 (**A**), Orai1 (**B**), and Orai3 (**C**) fluorescence intensity normalized by the number of nuclei in the mesenteric arteries with respect to endothelium-dependent vasodilation, determined as –log molar of concentration required to obtain 50% relaxation (pEC_50_) for bradykinin (BK) in isolated mesenteric arteries obtained from the same subject. Coefficients of determination (r^2^) and *p* values (in ***bold italic*** when significant) are indicated for each association. *n* indicates the number of subjects for the determinations.

**Figure 5 cells-11-03675-f005:**
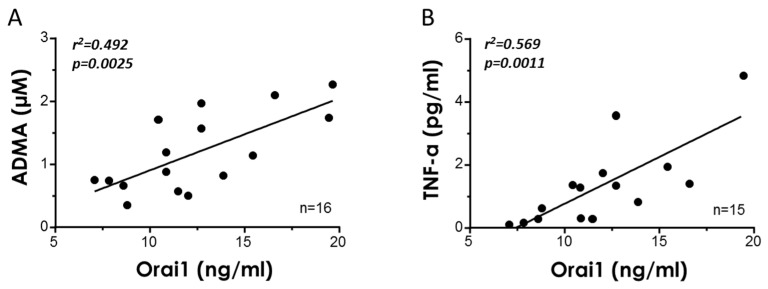
Plasma Orai1 concentrations are associated with biomarkers of endothelial dysfunction and inflammation. Simple regression analysis for plasma concentrations of Orai1 with respect to the plasmatic levels of asymmetric dimethyl arginine (ADMA) (**A**) and tumor necrosis factor-α (TNF-α); (**B**) determined in the samples from the same subjects. Coefficients of determination (r^2^) and *p* values (in ***bold italic*** when significant) are indicated for each association. *n* indicates the number of subjects for determinations.

**Table 1 cells-11-03675-t001:** Characteristics of the study subjects.

	<65-Years Old (*n* = 30)	>65-Years Old (*n* = 15)	*p*-Value
Age (years)	43.0 ± 10.8	73.7 ± 5.9	<0.001
Female (%)	7 (23.3)	7 (46.6)	0.172
Diabetes Mellitus (%)	1 (3.3)	4 (26.7)	0.042
Hypertension (%)	3 (10.0)	9 (60.0)	0.001
Dyslipidemia (%)	3 (10.0)	5 (33.3)	0.099
Cardiovascular Disease (%)	3 (10.0)	4 (26.7)	0.212

Age is expressed as mean ± standard deviation, while discrete variables are expressed as number and percentage (in parentheses).

## Data Availability

All data are available upon request to corresponding author.

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
