# Peer review of "Functional Role of STIM-1 and Orai1 in Human Microvascular Aging"

_cells, 2022, doi:10.3390/cells11223675_

Round 1

Reviewer 1 Report

Here the authors describe the role of STIM-1 and Orai1 channels in human vascular differences and aging comparison. The research is really nice, and important, but I have some comments on different small points that I would like to get clarified to improve this nice research presentation and support.

Major points

- At the introduction and at the discussion there should be some reference about how these calcium channels (homeostasis) could be changing Ca activated ion channels, why I say this, because for example Endothelial dependent Hyperpolarization responses are affected for channel opening interchange controlling the membrane potential and finally inducing changes in responses. 

- In Table 1 you showed that a percentage of patients are female subjects, have you performed statistical comparisons to see if these patients relate to the other male counterpart that you have in the groups?, please make a statistical evaluation of these, maybe you can put that comparison in the supplementary data, in order to be certain that you mixed them and present them as a one group as you are doing currently if differences are too strong, then Females should be separated in the presentation of the results, we do not know if menstruation phase/non-menstruation will have an impact in these channels expressed, which probably does.

- In the delimitation of patients there is no information about medications taken by these patients that could be related to calcium homeostasis, have you delimitated the inclusion of patients for patients not having antihypertensive, immunomodulatory medications, etc that could affect this calcium homeostatic balance? This is important for the appropriate design of your research.

-  Even so, there is immunofluorescence performed for expression, I do not see some basic staining (hematoxylin and eosin or trichrome ones) that could also tell about the layer's thickness and probable reorganization of fibers, which is an important point in hypertension, could some of that affect your results?, there is no description of that even so you are performing a comparison between ages, which could have changes related to it. 

- Please explain when the endothelial function was assessed in your samples. I do not think this was done after treatments?, You stated about the description of 50% relaxation (pEC50), but all samples relaxed the same way?, or did you delimit your endothelial presence to a 10% or an 80% of relaxation depending on the vascular tissue relaxed? to be sure about the endothelial function viability. I am asking because normally you will include some if they relax more than a certain value, but here I can not see this?

- From the previous, I can see some description in the mesenteric arteries, but not for the aortic samples endothelial evaluation. 

- The other main point is that you present normalization of the western blot quantification with Beta actin, but I can see that B-actin will variate too much from aging groups, meaning that this protein can not be used for normalization purposes as a housekeeping protein, please try another protein, or hopefully full protein quantification. If not, any can be used, I would prefer a comparison even with raw values, of course, the best is to find a way to normalize to an independent protein/variable that does not variate from aging changes. It would be nice to see the full blots at the supplementary data, please mark Kilo Dalton weight of the quantified bands, there is no description of it

 - About negative correlations, they seem to exist, but I do not believe this would be such important to make a correlation, values do not seem really good in terms of r2 values. It would be nice to present the data in another way. Maybe this variability is correlated with the females included or previous delimitations of inclusion and previous patients' treatments?

General things

- Please be certain to use MA or HMA all the time in one way, I have seen both ways to state this abbreviation. Some of the same are to check KHS and KKHS.

- In the abstract in line 27 "Orai3 was determined" should be "Orai3 were determined", you have several proteins evaluated not only one.

- In the introduction at line 52 "related to other vascular", should be related as another vascular".

- Please describe if you have performed antigen retrieval step in the immunofluorescence. I think there is a big jump here and more should be stated in the method description. 

- in line 223 you wrote "vascular vessel" and should be "vascular vessels"

- What about of the involvement of ORAI channels and bradykinin relaxations?, have you used YM-58483 to evaluate the endothelial function in this case?, would this be important in aging terms?

- Please use the same ranges in the graphs, for example Figure 1 A and B it would be nice to see them with a range from 0 to 300 so that we can compare how the responses change over age changes or the aging process that you are explaining along the article.

Reviewer 2 Report

In this manuscript, Assar et al. demonstrated the role of STIM/Orai in aging-related vascular alterations in human macro and microvasculature. They evaluated expression of STIM1, Orai1 and Orai3 in aorta and MA of younger and older than 65 years organ donors. They showed increased expression of Orai1 in human aorta and increased expression of Orai1 as well as STIM1 in MA of older donors. Authors further show that YM-58483 (Orai channel inhibitor) inhibited adrenergic-induced contractions in MA but had no effect on aging-related hypercontractility in aortic strips. They report inverse correlation of STIM1 and Orai1 protein expressions to endothelial function in MA. Overall, the study is good and novel.

In addition, there are a number of concerns listed below that should be addressed

1.     What is the status of Orai2 mRNA/protein in aorta and MA of younger and older than 65 years old?

2.     On page 5, line-192, MA is not dependent on the increased presence of hypertension or diabetes with aging.. Please show the data for the same.

3.     On page 10, line-299, Plasma Orai1 concentrations are associated with age-related circulating markers of endothelial dysfunction and inflammation. Please show the data for the same. Also show data with STIM1 as well.

4.     What is the status of STIM1, Orai1 and Orai3 protein expression in MA (western blot). Please show the data for the same.

5.     Please provide interpretation from the results and kindly discuss the conclusions from each figure.

6.     Throughout the manuscript, in figures please write figure panel on left side instead of right side.

7.     In figure 2 and 3, for western blot quantitation, please use dot plots instead of bar graphs to present the data.

8.     In figure 2 and 3, please add/provide scale bars for panels (E and F) and (A-F) respectively.

9.     Throughout the manuscript, analyses is probably meant to be “analysis”. Please correct this.

1.  NE should be defined and text should describe the concentrations and length of time of the treatment (Page 5 line-194 addition of cumulative concentrations of NE….).

1.  ADMA and BK should be defined in text.

1.  Throughout the manuscript, Ca2+ should be corrected to Ca2+.

Reviewer 3 Report

1.     In Figure 2E and 2F as well as Figure 3A-3F, the images are missing scale bars, please add them; the resolution of these images is also a little low, please adjust.

2.     It is not clear from the Figure 1A and 1B which two groups are comparing, please mark them with short lines on the images.

3.     In line 163, there is a redundant full stop after “Image J software”, please delete it.

4.     In line 392, there is a full stop missing after “vascular aging [38]”, please add it.

Round 2

Reviewer 1 Report

Thanks for your responses,

it would be nice to have actually the 300% scale as presented in your example late answer, since I think it would be easier for the reader to compare differences, even so is not the intent of the research/article but easier to digest and see in context. Significances are observed fine.

Nice research, I hope the best for the development in the research area and further ion channels association mechanisms. 

Author Response

Following reviewer's suggestion new Figure 1A has now 300% scale.

We thank the nice comment of the reviewer

Reviewer 2 Report

All my questions have been addressed. 

Author Response

We thank the comment of the reviewer